# Estrogen-Receptor-Positive Breast Cancer in Postmenopausal Women: The Role of Body Composition and Physical Exercise

**DOI:** 10.3390/ijerph18189834

**Published:** 2021-09-18

**Authors:** Ivan Dimauro, Elisa Grazioli, Cristina Antinozzi, Guglielmo Duranti, Alessia Arminio, Annamaria Mancini, Emanuela A. Greco, Daniela Caporossi, Attilio Parisi, Luigi Di Luigi

**Affiliations:** 1Unit of Biology and Genetics of Movement, Department of Movement, Human and Health Sciences, University of Rome Foro Italico, Piazza Lauro de Bosis 15, 00135 Rome, Italy; daniela.caporossi@uniroma4.it; 2Unit of Physical Exercise and Sport Sciences, Department of Movement, Human and Health Sciences, University of Rome Foro Italico, Piazza Lauro de Bosis 15, 00135 Rome, Italy; elisa.grazioli@uniroma4.it (E.G.); attilio.parisi@uniroma4.it (A.P.); 3Unit of Endocrinology, Department of Movement, Human and Health Sciences, University of Rome Foro Italico, Piazza Lauro de Bosis 15, 00135 Rome, Italy; cristina.antinozzi@uniroma4.it (C.A.); alessia.arminio@virgilio.it (A.A.); emanuela.greco@unicz.it (E.A.G.); luigi.diluigi@uniroma4.it (L.D.L.); 4Unit of Biocheminstry and Molecular Biology, Department of Movement, Human and Health Sciences, University of Rome Foro Italico, Piazza Lauro de Bosis 15, 00135 Rome, Italy; guglielmo.duranti@uniroma4.it; 5Dipartimento di Scienze Motorie e del Benessere (DISMeB), Università Degli Studi di Napoli “Parthenope”, Via F. Acton, 38, 80133 Naples, Italy; annamaria.mancini@uniparthenope.it; 6CEINGE-Biotecnologie Avanzate s.c.ar.l., Via Gaetano Salvatore 482, 80145 Naples, Italy; 7Department of Health Science, University “Magna Graecia” of Catanzaro, Viale Europa, 88100 Catanzaro, Italy

**Keywords:** breast cancer, physical activity, body composition, risk factors, cancer prevention

## Abstract

Breast cancer (BC) is the most commonly diagnosed cancer among women worldwide and the most common cause of cancer-related death. To date, it is still a challenge to estimate the magnitude of the clinical impact of physical activity (**PA**) on those parameters producing significative changes in future BC risk and disease progression. However, studies conducted in recent years highlight the role of **PA** not only as a protective factor for the development of ER+ breast cancer but, more generally, as a useful tool in the management of BC treatment as an adjuvant to traditional therapies. In this review, we focused our attention on data obtained from human studies analyzing, at each level of disease prevention (i.e., primary, secondary, tertiary and quaternary), the positive impact of **PA**/exercise in ER+ BC, a subtype representing approximately 70% of all BC diagnoses. Moreover, given the importance of estrogen receptors and body composition (i.e., adipose tissue) in this subtype of BC, an overview of their role will also be made throughout this review.

## 1. Introduction

Breast cancer (BC) is the most commonly diagnosed cancer among women in 140 of 184 countries worldwide, and it is the most common cause of cancer-related death in 103 countries [1]. To date, it is considered curable in ~70–80% of patients with early-stage, non-metastatic disease [2].

BC is known to be a hormone-dependent disease characterized by molecular mechanisms involving activation of human epidermal growth factor receptor 2 (HER2, encoded by ERBB2), hormone receptors (estrogen receptor and progesterone receptor) and/or BRCA mutations [2]. Most BCs (70–80%) express a significant amount of estrogen receptors (ER) and/or progesterone receptors (PR), which are considered biomarkers of a favourable prognosis [3].

Although BC is defined as a malignant tumor that affects the breast, there are recognizable different types on the base of specific breast cells involved. The following classification is made according to the stage of the tumor and where it takes place. In particular, it is possible to distinguish the ductal carcinoma in situ (DCIS), which is considered non-invasive or pre-invasive, whose cells have become cancerous but have not yet invaded the surrounding tissues, and nonobligate precursors of invasive BC (20% of screen detected), as well as the invasive BC whose cancerous cells have reached the surrounding tissues [1,4]. It is estimated that about 50% of DCIS patients will progress to invasive cancer [4].

Another classification is made according to the molecular subtype, determined by the analysis of the gene expression of HER2, and by quantitative hormone receptor (HR) analysis [5], which so far has identified four main subtypes: (1) Luminal A (HR+/HER2^−^), the most common type that tends to be slower growing and less aggressive with the most favourable prognosis; (2) Luminal B (HR+/HER2+), which results in HR+ and is highly positive for Ki97 and/or HER2 protein with poorer outcomes; (3) Basal-like (HR−/HER2−), also called triple-negative because of ER−, PR− and HER2−, with the worst prognosis of all other subtypes and a very low survival expectancy; (4) the HER2-enriched (HR−/HER2+) [6] (Table 1).

To date, women with a history of BC represent the largest group of cancer survivors in high-income countries [7]; thus, it becomes a category of patients who require an increasingly demanding management.

The implementation of an effective intervention plan is necessary at each level, from the origins of the cause to the management of disease; therefore, it is a priority to understand the risk factors leading to the development of disease and possible interventional approaches.

In this narrative review, we focused our attention on data obtained from human studies analyzing, at each level of disease prevention (i.e., primary, secondary, tertiary and quaternary), the positive impact of physical activity (**PA**)/exercise in sedentary/non-active subjects (<2 h × week) with ER+ BC, a subtype representing approximately 70% of all BC diagnoses. Moreover, given the importance of ER and body composition (i.e., adipose tissue) in this subtype of BC [8], throughout this review an overview of their role will also be made.

## 2. Estrogen Receptors in Breast Cancer and Their Clinical Implications

Some of the features of human BC (e.g., initiation and progression) are derived from a deregulation of estrogen-dependent and ER signaling pathways [9] (Figure 1).

It is known that the effects of estrogen are mediated by three different ERs: (1) the nuclear receptor ERα, which drives almost ∼75% of BCs [10]; (2) nuclear receptor ERβ; (3) the cytoplasmic G protein-coupled estrogen receptor 1 (GPER) [11,12,13,14].

ERα and ERβ share common structural characteristics with five different domains, named A/B, C, D, E and F, with similar mechanism of action [14,15]. Generally, estrogens move the cell passively by diffusion through the cellular membrane, bind ERs in the cytoplasm and are transported to the nucleus [16]. The interaction receptor-ligand induces conformational change of the receptors, whereby the ERs form dimers, bind DNA and initiate gene transcription [14]. In this case, ERs regulate transcriptional processes by nuclear translocation and binding to specific response elements, which act on the regulation of gene expression [17].

Three ERα isoforms have been identified in mammals: full-length ERα, and two truncated isoforms, ERα36 and ERα46, respectively. ERα36 expression has been particularly detected in BC, as well as in endometrial, colorectal, gastric and hepatic cancers [14,18].

ERα and ERβ are distributed differently in human breast tissues: ERα expression is mainly limited to the nuclei of epithelial cells present in the lobules and ducts of the healthy breast. Differently, ERβ is also expressed in normal breast tissue, where it is detectable in myoepithelial cells as well as in surrounding stromal and endothelial cells [14,19,20].

BCs not expressing ERα were tested positively for ERβ expression. It has been demonstrated that ERβ possesses a weaker activity than ERα, able to repress the transcriptional activity of ERα, although this mechanism remains to be investigated [14,19,21]. A comprehensive clarification about the role of ERβ in BC is hampered by the presence of five different isoforms of ERβ (ERβ, β2, β3, β4, and β5). However, although more investigations are needed, the general consensus is its suppressor role in BC, since it is able to reduce growth, proliferation and cancer cell migration and invasion mediated by ERα, [14,22,23,24]. Besides its genomic actions, ER mediates non-genomic effects towards the transmembrane protein, GPER, commonly accepted as being responsible for the extra-nuclear, non-genomic effects of estrogens [14,25].

Multiple ER-targeting drugs are used routinely in the clinical practice to treat ER+ BC patients; however, initial or acquired resistance to these therapies frequently occurs, with recurrence of metastatic tumors [9]. Therefore, understanding the mechanisms leading to drug resistance becomes extremely important. In normal conditions, the activity of ER is controlled mainly by the availability of estrogens, which bind the ER-ligand-binding domain and mediate receptor dimerization, nuclear translocation and the binding to estrogen response elements (EREs), located close the promoters of target genes [21]. Different studies demonstrate that growth factors, hormones and cytokines produced by the tumor microenvironment play pivotal roles in the progression of ER+ BC, and many of these signaling pathways can directly affect the transcriptional activity and function of ER, independently by the classical estrogenic ligands. In particular, the ER phosphorylation may have a key role in the receptor activation in a ligand-independent manner [26].

Phosphorylation of Ser118 (S118) is one of the most well-characterized systems of ER activation independently of estrogens and can be induced by epidermal growth factor (EGF) and mitogen-activated protein kinase (MAPK) [27,28,29]. This EGF-induced phosphorylation has been demonstrated to be involved in increasing cell proliferation in tumorigenic cells, favouring the binding of ER to chromatin through cooperation with several transcription factor complexes, such as AP-1 transcription factors and early B-cell leukaemia transcription factor 1 (PBX1) [30,31]. To be noted, the Y537 and D538 ER BC mutants are constitutively phosphorylated on S118 in an estrogen-independent manner [32,33], highlighting the importance of this phosphorylation event for ER activity and suggesting the S118 as a fundamental regulatory site in the drug-resistant metastatic disease. Phosphorylation of S305 appears to be important for this estrogen-independent activation of ER as well. This event is mediated by the protein kinase-A (PKA) [34] and Pak1 [35] in the absence of estradiol and drives receptor activity that is refractory to tamoxifen inhibition [34].

Particularly, PKA-mediated ER phosphorylation induces receptor binding to non-classic regulatory sites that differ from those typically bound by ER after estradiol-induced activation. It was suggested that this mechanism induces the expression of the oncogene c-MYC responsible of tamoxifen resistance [36]. The phosphorylation on S305 can also be induced by inflammatory molecules and adypokines as leptin, TNF-alpha, IL6 and IL1-beta, produced by different cell types involved in cancer progression [37]. The cytokine-induced phosphorylation is mediated by the inhibition of nuclear factor κB kinase subunit β (IKKβ), rather than PKA or Pak1, and it is involved in cell extravasation, an important part of the metastatic process [37]. Once ER binds chromatin, other transcription factors can redirect the binding on DNA, reprogramming the transcriptional activity of ER to other target genes. These factors include FOXA1 [38], PBX1 [31], the transcription factor AP-2γ [39], and GATA-binding protein 3 (GATA3) [40]. Therefore, different pathways triggered by molecules, produced by the tumor microenvironment, can impact ER function and influence endocrine resistance. This highlights the need to understand as much as possible the molecular mechanisms related to factors involved in ER+ BC.

To date, few studies performed in rat models following physical training analyzed the expression of ER in BC cells. In these studies, authors demonstrated an effect of **PA** in increasing the ratio of ERβ/ERα, and a reduction in the sensitivity of BC cells to the pro-proliferative and antiapoptotic effects of estrogen, leading to apoptotic cell death [41].

However, further studies in human and in disease-applicable preclinical models could be useful to validate these mechanisms and determine if these pathways may provide molecular tools for therapeutic application.

## 3. The Role of Adipose Tissue

According to the anatomical location and to the main cell component, adipose tissue can be divided in three different types: white adipose tissue (WAT), which represents more than 95% of the fat mass, brown adipose tissue (BAT), which constitutes 1% to 2% of fat, and the most recently discovered beige adipose tissue [42].

This tissue derives from WAT by a conversion process called the browning of adipose tissue but resembles BAT in morphology and role. The formation of beige adipocytes is reversible and is generally a consequence of adrenergic stimulation, cold exposure, diet and exercise [43,44,45].

The WAT is the main storage site for energy deposition, and it is composed of mature adipocytes capable of storing energy in the form of triacylglycerol (TAGs) in lipid droplets. Only 20–30% of adipose tissue is made up of mature adipocytes; the remaining 70–80% is composed of the stromal vascular fraction (SVF) [46,47], connective tissue matrix, as well as vascular and neural tissues. The non-adipocyte cellular component includes various types of immune cells such as macrophages, neutrophils, eosinophils, mast cells, lymphocyte T cells and B cells, as well as preadipocytes and fibroblasts [48,49].

This cellular heterogeneity clearly demonstrates that adipose tissue is a complex organ with different functions, regulating the metabolism of the whole body [47]. In particular, it is considered an endocrine organ releasing numerous substances, such as adipokines (i.e., adiponectin, leptin and resistin), hormones, as well as cytokines (i.e., TNF-α, IL-6, IL-1, IL-8) [50]. This explains the excess adipose tissue in the body, which contributes to the onset of a pathological state of many organs and systems [51]. Among the different types of adipose tissue, WAT is responsible for the inflammation process. This means that the inflammation grade increases with the progression of obesity, and the results strongly associated with an increase in adipocytes size and a systemic insulin resistance (IR) state.

Numerous preclinical and clinical studies demonstrated that chronic low-grade inflammation of adipose tissue, also called “metaflammation”, is strongly and consistently associated with excess body fat mass and metabolic disease onset and progression. This mechanism is initiated and sustained over time by adipocyte dysfunction, which releases inflammatory adipokines, and by the infiltration/activation of numerous immune cells (i.e., pro-inflammatory M1 macrophages, dendritic cells, mast cells, neutrophils, B cells, and T cells), which amplify the inflammatory response through the production/secretion of proinflammatory cytokines and chemokines [52,53]. Differently from the acute inflammation, metaflammtation is characterized by chronic low-grade inflammation, since cytokine release and immune cell infiltration come out gradually and remain unresolved over time [52,53].

Besides adipokines and inflammatory mediators, adipose tissue is responsible for estrogen production. Especially in postmenopausal woman, the increase in **BMI** is associated with a high release of estrone, estradiol and free estradiol, as well as a high expression of the enzyme aromatase. These abnormal changes lead to excessive, 10-fold estrogen secretion in the breast and therefore to a higher risk of developing BC [54,55].

As shown in Figure 2 adipocyte hypertrophy leads to “unhealthy” adipocytes characterized by mitochondria dysfunction, which produces reactive oxygen species (ROS), lipolysis and insulin resistance. The inability of insulin to suppress lipolysis increases free fatty acid (FFA) mobilization. In the absence of FFA utilization, they can trigger adipocyte inflammation and increase inflammatory macrophages (M1), which also produce TNF-α, a cytokine able to induce a supra-physiological production of ROS through the inhibition of insulin signaling and mitochondrial function.

Exercise training has been shown to have a multitude of health benefits, including those at a metabolic, antioxidant, and anti-inflammatory level [56,57,58,59,60]. For instance, resistance training positively affects WAT metabolism. Indeed, exercise decreases blood glucose levels and increases the activity of different hormones, including glucagon, catecholamines (epinephrine and norepinephrine), growth hormone (GH), atrial natriuretic peptide (ANP), brain natriuretic peptide (BNP) and cortisol. These molecules act as lipolytic hormones and regulate the release of FFA and glucagone, providing energy substrates for skeletal muscle cells [61,62,63]. By creating a negative energy balance, fat loss is facilitated, which occurs as a first adaptation with the reduction in the adipocytes size, which becomes more insulin-sensitive, a mechanism by which the inflammation of the WAT and the dysregulated lipolysis are reduced. Moreover, the stimulation of lipolysis and insulin sensitivity following exercise training is correlated with increased FFA oxidation and lower lipid storage in WAT.

Other important effects related to training concern inflammation of the adipose tissue and mitochondrial function. In fact, the adaptation process involves not only the mitochondria in skeletal muscle, but also those present in adipose tissue. A higher density of mitochondria is characteristic within BAT, but mitochondrial biogenesis can be induced by exercise training, improving the brown adipocyte-specific gene expression and the phenotypic switching from WAT to BAT. This framework of training-induced adaptation clarifies not only the importance of weight loss, but also better adipose tissue function through increased mitochondrial activity and a reduced inflammation level. This makes the fat cells “fit”, which can be used in favour of a healthy body [64].

## 4. Physical Activity and Physical Exercise in Breast Cancer Prevention

Since the interchangeable use of exercise and **PA** in research literature, Dasso [65] tried to clarify the differences. The author reported the Center for Disease Control and Prevention definition of exercise, which is a subcategory of **PA**, intended as a planned, structured, repetitive, and purposive activity so that the improvement or maintenance of one or more functional parameters can be objectively quantified. Instead, as reported by the World Health Organization [66], **PA** consists in any bodily movement produced by skeletal muscles that result in energy expenditure. According to the author, a clear definition of exercise allows health care providers to speak to patients about improving their **PA**.

However, regardless of the definitions, it is known that both **PA** and exercise play a pivotal role on health status, helping to prevent different diseases, including cancer [56,57,67,68,69,70,71,72,73,74,75,76,77,78,79,80,81].

To date, five levels of disease prevention are recognized [82]:***Primordial prevention***—It consists in programs and campaigns, usually addressed to the younger population, aimed at promoting a healthy lifestyle and avoiding the incurrence of risk factors;***Primary prevention***—It consists in measures, addressed to a susceptible but healthy population, aimed at preventing a disease through specific activities that limit risk exposure or increase the immunity;***Secondary prevention***—It consists in procedures that increase the early disease detection, and its target is healthy-appearing individuals with subclinical forms of the disease, and often occurs in the form of screenings. The objective is the early identification of sick or high-risk subjects to achieve healing or prevent the onset and progression of the disease;***Tertiary prevention***—It targets both clinical and outcome stages of a disease, with the aim to reduce the severity of the disease as well as of any associated consequences, and to reduce the effects of the disease once established in an individual, through a tailored rehabilitation program;***Quaternary prevention***—It consist in practice able to protect patients from medical interventions that cause more harm than benefits, due to the over-treatment condition or final-stage of the disease.

Epidemiological studies have highlighted a variety of modifiable and non-modifiable BC risk factors [83], also known as host and environmental factors [84,85].

Among the non-modifiable risk factors, there are: family history of the cancer, BRCA1 and/or BRCA2 mutations, reproductive factors that influence endogenous estrogen exposure (nulliparity, early age at menarche, later menopause, and later age at first full-term pregnancy), race and ethnicity.

Differently, the modifiable risk factors include alcohol drinking, physical inactivity, excess body weight, as well as the use of exogenous hormones (oral contraceptives and menopausal hormone replacement therapy), and smoking [83].

BC is considered a heterogeneous disease with 30% of cases recognized as familial BC (which reveals the association with number of high-, moderate-, and low-penetrance susceptibility genes) compared to 70% of cases of BC presenting as sporadic [86]. These data highlight the importance to modify those risk factors correlated with the lifestyle to potentially reduce the disease incidence [87].

Particularly interesting are the results of several studies, which found a strong correlation between adipose inflammation and an alteration in estrogen biosynthesis/signaling pathways in obese patients. The excessive chronic exposure to estrogens increases the risk of developing BC [11,88,89]. This led to the recognition of obesity as a risk factor for the hormone-dependent subtype of BC, especially in postmenopausal women [15].

**PA** has been demonstrated to positively impact specific biomarkers related to physiological and pathological condition [56,57,75,76,90,91,92,93,94,95,96,97] or to reduce the incidence of cardiovascular and metabolic diseases in broad populations of individuals, including women, older individuals, patients with coronary heart diseases [98], as well as those with diabetes [77,97,99] and heart failure [98]. Moreover, numerous research studies have identified the **PA** as an important factor in the primary prevention of BC able to ameliorate the patient conditions in the different stages of the disease, either after the diagnosis of BC or in the early post-surgery steps improving survival outcomes [67,68,70,71,72,73,100]. Indeed, the authors have been demonstrated positive effects of well-structured aerobic or strength training protocols on Quality of Life (QoL), fatigue, aerobic fitness, and muscular strength in BC survivors during and after treatments [101,102,103]. Moreover, it is known that the beneficial association between **PA** and BC survival are partially related to biological and biochemical changes capable of influencing several hormones (i.e., sex hormones, insulin, **IGF-1**) and DNA methylation levels of specific tumor suppressor genes, which appear to be directly involved in the progression of this disease [104].

## 5. Physical Activity and Exercise as Fundamental Approaches within ER-Positive Breast Cancer at Each Disease Prevention Level

The strong association between **PA**/exercise and BC risk is widely recognized, reporting beneficial effects on tumor number sites, growth, metastasis and incidence [105,106]. The mechanisms involved in these processes are complex and multifaceted and may be mediated, at least in part, by a reduction in inflammation markers, particularly MCP-1 and IL-6 [105], and the reduced expression of the transcription factor NF-κB [107]. To date, most of the main findings regarding the molecular mechanisms involving physical exercise and tumorigenesis were obtained in preclinical models available elsewhere [108,109,110,111,112].

In the next paragraphs, we have reported data from human studies related to the impact of **PA** on BC at each level of disease prevention: primary, secondary, tertiary and quaternary. In particular, we focused on studies analyzing whether or not the amount of exercise prescribed influences BC biomarker levels in patients with a sedentary lifestyle (<120 min/week of moderate-to-vigorous **PA**) [113].

Unfortunately, no studies are still conducted on the possible role of physical exercise as “primordial” BC intervention.

### 5.1. Primary Prevention

The American Cancer Society (ACS) and the American Institute for Cancer Research/World Research Fund (AICR/WCRF) publish **PA** guidelines for cancer prevention. For an adult, the ACS guideline recommended 150 min/week of moderate activity or 75 min/week of vigorous activity throughout the week [114]. The AICR/WCRF recommends 30 min/day of moderate activity increasing to 60 min/day of moderate activity or 30 min/day of vigorous activity as fitness improves [115]. However, to date, questions remain about the applicability of these guidelines with respect to reductions in BC risk.

Furthermore, even if an exercise intervention achieves these targets, it is unknown what the long-term implications are for postmenopausal BC risk.

Previous randomized controlled trials [116,117,118,119,120,121,122,123,124,125,126,127,128,129] have highlighted several plausible biological mechanisms whereby **PA** can reduce postmenopausal BC risk (Table 2).

These studies presented in Table 2 highlight how **PA**/exercise interventions contribute to the modulation of adiposity, endogenous sex and metabolic hormones, and inflammatory markers. A careful analysis of the research papers showed that these effects, especially those on sex hormones, were dependent on the characteristics of exercise (i.e., type, intensity, duration and frequency) and on the homogeneity of the subjects recruited in terms of **BMI**. In particular, it was evident that 12 months of training (5 d/w, 45 min at 60–85% **HRmax**), mainly of endurance activity, in subjects with a narrow range of **BMI** between overweight and obesity, showed a significant decrease in fat mass, free and total testosterone, estrone, free and total estradiol, as well as an increase in **SHBG** [117,119,126,129].

These results seem to be achieved when people also followed a healthy dietary lifestyle combined with training, evidencing a decrease in insulin, adiponectin, leptin and hsCRCP levels, likely associated with a lower BC risk [119,120,121,123]. A change of exercise characteristics such as type and frequency, as well as the choice of a non-homogeneous sample, was sufficient to determine a lower impact of **PA** [116,122,127]. A minor impact of **PA** was observed when the training period was shortened, and the subjects recruited had a wider range of BMI [121].

Interestingly, in 2019, both Duggan et al. [123] and Friedenrich et al. [118] demonstrated that most of the beneficial effects of **PA** were maintained even after 18 months of follow-up. Still, the caloric restriction combined with exercise seems to be most beneficial for lowering sex hormone levels [119,123]. Comparing the combination of exercise and caloric restriction with caloric restriction only, all results favored the combination, even when weight loss between the groups was comparable. An additional important advantage of combining caloric restriction with adequate protein intake and regular physical activity is the preservation of as much muscle mass, and thereby muscle strength, physical function and cardiovascular fitness, as possible [130,131,132].

Finally, Gonzalo-Encabo et al. [124] and van Gemert et al. [120] demonstrated that a combined exercise (i.e., resistance and endurance activity), adapted from the American College of Sports Medicine’s guidelines [133], improves body composition and sex hormone profile in postmenopausal women, known to be important risk factors for ER+ BC.

### 5.2. Secondary and Tertiary Prevention

The latest evidence supports the role of exercise prescription to reduce morbidity, improves function and quality of life, and potentially improves survival, with very low risk of harm [67,134,135]. As suggested by Schmitz [136], on the basis of other diseases’ rehabilitation programs, a model to integrate exercise prescription into cancer clinical care is needed to reduce the risk of mortality and recurrence, through improvements in functional capacity, body composition and other several factors [137].

To date, numerous side effects related to adjuvant hormone therapy were reported, such as biological (dyslipidemia), physical (weight gain, hot flashes, vaginal dryness, sexual disorders with low libido, musculoskeletal alterations), and psychosocial (anxious-depressive disorders, poor body image, difficulties of professional reintegration) [138,139].

More specifically, the aromatase inhibitor, a drug commonly prescribed for postmenopausal ER+ BC, has been shown to induce an increase in body fat and a reduction in insulin sensitivity [140,141,142], as well as long-term cardiotoxicity [143,144,145], osteoporosis [146,147,148], and arthralgia [149,150,151].

In this section, we included trials proposing exercise and **PA** protocols for BC patients (from I to III stage) before, during, or after the classic pharmacological treatments, when they reported the ER + or the aromatase inhibitors in the patient characteristics (Table 3).

Shmitz et al. [152] evidenced that weight exercise protocols can be performed safely and without contraindications immediately after the traditional cancer treatments, increasing muscle mass, as well as decreasing body fat % and IGF-II levels. During recent years, literature showed the effectiveness of training, not only after the treatment, but also during radiotherapy and chemotherapy [157,165,166]. One of the first studies on ER+ patients compared aerobic and strength protocols performed at home [167], with a no intervention group; data reported a most evident increase in aerobic capacity and a reduction in body mineral density (BDM) decline in the aerobic group compared to the others, suggesting that home-based aerobic protocol may prevent or at least minimize bone loss observed during chemotherapy, counteracting the long-term side effects. Similarly, Ligibel and colleagues [154] highlighted, in BC patients undergoing hormonal therapy (**HT**), the effects of a combined protocol, unsupervised aerobic and supervised strength training, on fast insulin levels and hip circumference, despite no insulin resistance, fasting glucose or body mass index (**BMI**) modification being reported. These results support the relationship between **PA** and BC prognosis through the modulation of insulin levels and/or body fat or fat deposition. A similar type of protocol, supervised + unsupervised aerobic activity, was proposed to BC after **CH** [156], and the results confirm that a moderate-intensity aerobic exercise can induce favorable changes in body composition improving the disease prognosis. Moreover, in recent years, studies have focused their attention on the effect of exercise on C-reactive protein (**CRP**) and interleukins in BC patients [157,159,160,167]. The aerobic protocol proposed by Guinan et al. [157] showed a decrease in waist circumference, but no changes were evidenced in **CRP**, blood pressure, high-density lipoprotein (**HDL**), and insulin resistance, probably because 8 weeks are not enough to produce these types of modulations. On the other hand, Rogers et al. [160] proposed a combined protocol, unsupervised aerobic and supervised strength, to 15 BC patients to evaluate the effects on the inflammatory system. The results showed an improvement in predicted oxygen consumption and sleep latency, but, contrary to the hypothesis, an increase in IL-6 and a decrease in IL-10 and adiponectin, probably due to the small sample size or to the differences between the group baseline characteristics. The following year, the same authors [159] confirmed the previous results on IL-10, sleep dysfunction and **VO_2max_**, showing a decrease in % of body fat in the active group. They also revealed that the increase in fatigue intensity seems to be mediated by interleukin IL-6 and IL-10; instead, the decrease in fatigue interference could be mediated by sleep dysfunction. Moreover, the reduction in general fatigue could be mediated by minutes of **PA**, sleep dysfunction, and **PA** enjoyment. These results add significant data about the importance of biobehavioral factors as mediators of fatigue management in BC patients. According to de Paulo et al. [163], 9 months of supervised high-intensity combined training performed three times per week can increase those parameters, such as total and fat mass, **HDL** and osteocalcin levels, decreasing the side effects of aromatase inhibitors in old BC patients. In 2018 and 2019, Mijwel et al. [69,100] proposed two different types of high-intensity interval training (HIIT), as well as a combined and aerobic protocol, to BC patients during and after treatment. The first study evidenced a decrease in fatigue sensation in both active groups and a decrease in pain perception in the control group [164]. The second study [69] confirmed those results and showed an increase in muscle fiber cross-sectional area and satellite cell count per fiber in the combined HIIT group, an increase in the number of capillaries per fiber in the aerobic HIIT group and a decrease in **MHC** isoform type I and protein levels of PINK1 in the control group. These results illustrate the importance of exercise in patients undergoing chemotherapy to prevent the negative side effects of treatment and inactivity through preserving skeletal muscle mass and function. In conclusion, supervised aerobic and strength training performed during chemotherapy seems to be one of the best adjuvant treatments able to improve survivorship in ER+ BC patients [161]. Moreover, the combination between exercise and diet seems to be a new effective strategy to counteract the treatment side effect of the disease and treatments [155,158,162]. According to Demark-Wahnefried et al. [155], the combination of a calcium diet, 6 months of exercise and a fruit and vegetable, low-fat diet can decrease waist circumference and % body fat despite no change in insulin, proinsulin, **IGF-1**, **CRP**, cholesterol, **SHBG**, **IL-1B**, and **TNFR2** levels being detected. As reported by Scott et al. [158], an aerobic and strength protocol combined with a daily caloric reduction of 600 kcal below the patient’s requirement produces a decrease in central adiposity, waist to hip ratio (WHR), total cholesterol and leptin levels as well as an increase in predicted **VO_2max_**. These findings suggest that a well-tailored protocol combined with a hypocaloric diet positively impact on long-term prognosis in overweight BC patients undergoing treatments. Indeed, as reported by Artene et al. [162], diet is effective for ER+/PR±/HER2- BC patients on anti-estrogenic medication, but adding at least a minimal exercise protocol improves patients’ chances of counteracting sarcopenic obesity, fatigue, and other negative cancer-related effects.

### 5.3. Quaternary Prevention

The role of physical exercise is as essential in the quaternary prevention phase as it is in the secondary and tertiary prevention phases. It seems that, in patients with advanced cancer, exercise provides the maintenance and improvement of fitness and physical function and may diminish fatigue sensation, improving quality of life (QoL). It should be considered as an intervention to prevent further health complications [168,169]. Few studies investigated these effects related to specific types of cancer, which are often poorly described. Thus, in this section, the feasibility and the effect of exercise in patients at the terminal stage of BC are evidenced in the presence of metastases, under treatment, with a life expectancy of at least 4 months (Table 4).

According to the studies, the training protocols feasibility and effect on the metastatic BC patient undergoing treatments are still controversial [171,172,173,175]. It seems that an aerobic exercise, supervised or unsupervised, that reaches 150 min of moderate-intensity aerobic training per week is safe but not feasible [170], and it did not show significant improvements in physical functioning in a heterogeneous group of women living with advanced BC [171]. On the contrary, 8-week home-based supervised resistance training combined with an unsupervised walking program seems to be feasible, improving fatigue sensation, **VO_2max_** and functional capacity [173]. These data suggest that further studies are needed to explore alternative interventions to determine whether exercise could help women with metastatic disease to live more fully with fewer symptoms from disease and treatment. As highlighted by Carson et al. [170,174], **PA** such as yoga could be useful for the treatment of side effects. In particular, 8 weeks of meditation, gentle postures and breathing techniques seems to reduce the daily pain and fatigue distress, increasing relaxation and invigorating the patients.

Our analysis of the literature is synthesized in Figure 3, highlighting that the most beneficial effects of exercise for primary prevention of ER+ BC were found with both an endurance activity (e.g., 5 d/w, 45 min at 60–85% HR max) and a combined activity (i.e., 4 h/w endurance at 60–90% HR max, and strength at 65% of **1RM**); regarding secondary and tertiary prevention, the combination between endurance and strength training (i.e., 3 d/w, 60 min endurance at 80% **VO_2max_**, and strength at 70% of **1RM**) can affect the tumor development and progression during and after treatments; finally, for the quaternary prevention, 150 min per week of aerobic activity (e.g., yoga or walking activity) seems feasible and increases QoL in BC patients.

## 6. Conclusions

BC is one of the most common cancers in the world, but cancer survivors in high-income countries have also grown thanks to medical advances. Most BCs express ER and/or PR. Among the modifiable risk factors involved in excessive estrogen exposure, body fat was identified to be responsible for estrogen biosynthesis and their signaling pathways. Indeed, during the last decade, an increased prevalence of female obesity characterized by an unhealthy body composition was reported worldwide. It is precisely this group of women who are considered to be at a high risk of developing this BC subgroup.

To date, it is still a challenge to estimate the magnitude of the clinical impact of **PA** on the observed levels of sex hormones, since there are no absolute cut-off values defined that correspond with a certain change in future BC risk and disease progression. However, besides an important effect on circulating sex hormone levels, it should also be considered that different studies on pre-clinical BC animal models demonstrate an important effect of **PA** to increase the ERβ/ERα ratio, as well as an increase in mammary gland cell apoptosis [41,176,177]. 

Furthermore, studies conducted in recent years highlight the role of **PA** not only as a protective factor for the development of ER+ BC but also as a useful tool in the management of BC treatment as an adjuvant to traditional therapies.

In the future, it is a priority to introduce **PA** in health care paths as well as to improve our knowledge with studies based on exercise-related health promotion. We also hope to focus on the primordial prevention of BC to avoid, where possible, the development of risk factors in the first place.

## Figures and Tables

**Figure 1 ijerph-18-09834-f001:**
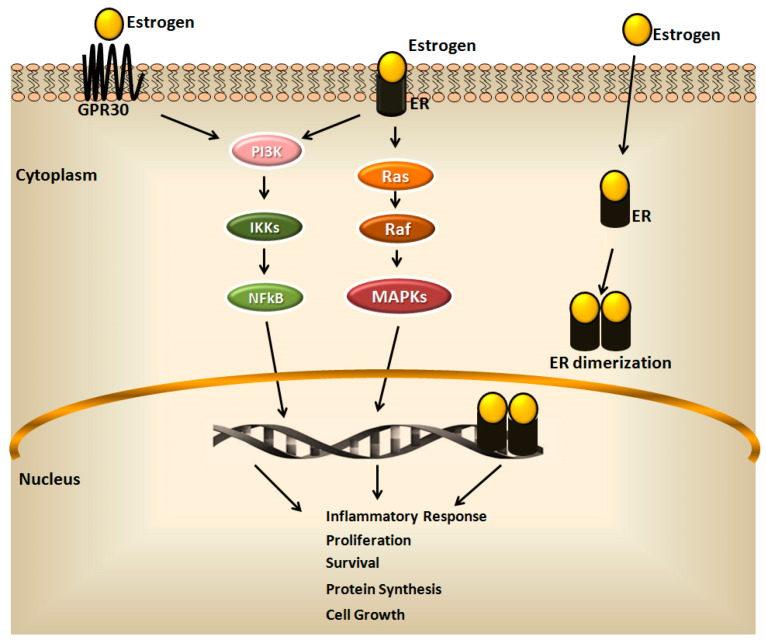
The estrogen signaling pathway. The estrogen signaling mainly includes activation of intracellular estrogen receptor (ER) that, upon ligand binding and dimerization, translocates to the nucleus, where it directly binds responsive elements of target genes involved in the cell growth, inflammation, proliferation, survival, and protein synthesis. Differently, estrogens mediate non-genomic effects and activate intracellular signaling through the binding of the plasma membrane receptors, ER variants and the G protein-coupled receptor (GPR30). This binding induces the rapid activation of protein kinases, phosphatidylinositol-3-kinase (PI3K), renin-angiotensin system (Ras) and rapidly accelerated fibrosarcoma (Raf), as well as the transcription factors nuclear factor kappa-light-chain-enhancer of activated B cells (NFκB) and mitogen-activated protein kinase (MAPK), which regulate the gene expression of estrogen target genes.

**Figure 2 ijerph-18-09834-f002:**
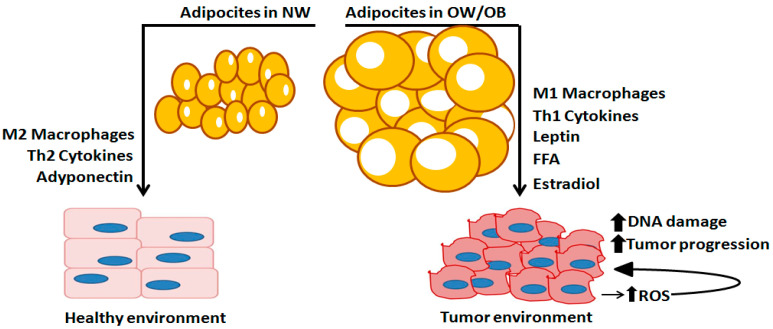
Adipose-related factors engaged to the initiation and progression of breast cancer. Normal adipose tissue, generally existing in “normal weight” (NW) subjects (%BF cutoff values between 8 and 20% for men and 14 and 23% for women), is characterized by smaller and less adipocyte cells and M2-polarized macrophages that release anti-inflammatory cytokines and adiponectin that contribute to normal breast cell development. In overweight (OW) and obese condition (OB), adipose tissue is characterized by larger size and more abundant adipocytes, releasing pro-inflammatory cytokines, M1-polarized macrophages, leptin, free fatty acids (FFAs), and estrogens, synthesized by aromatase enzyme. These factors act as mutagens stimulating the growth of tumor cells. A tumor environment produces more reactive oxygen species (ROS), which generate DNA damage, amplify and induce mutagenesis, tumor growth and progression. Therefore, the OW/OB condition provides a favorable microenvironment for adipose tissue to induce tumor establishment and progression.

**Figure 3 ijerph-18-09834-f003:**
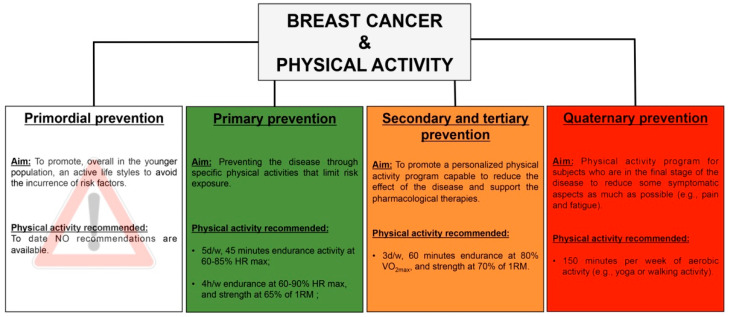
Physical activity recommendations based on scientific evidence and stratified for each prevention level of breast cancer. d, day; w, week; HR max, heart rate max; **1RM**, one-repetition maximum.

**Table 1 ijerph-18-09834-t001:** Molecular subtypes of breast cancer.

Subtypes	Molecular Signatures	% Incidence
**Luminal A**	ER+, PR±, HER2−, Low Ki67	≈70%
**Luminal B**	ER+, PR±, HER2±, High Ki67	10–20%
**Triple Negative**	ER−, PR−, HER2−	15–20%
**HER2**	ER−, PR−, HER+	5–15%

**Table 2 ijerph-18-09834-t002:** Exercise and physical activity in primary prevention of breast cancer in overweight/obese postmenopausal women.

REF.	GROUPS	EXERCISE CHARACTERISTICS	MAIN OUTCOMES
[128]	(*n* = 173) age 50 to 79 years **BMI** ≥ 25 kg/m^2^ (mean 30.4 ± 4.1) Groups: **AG** (*n* = 84) **CG** (*n* = 86)	12 months **AG**: endurance exercise, 5 d/w progressively increase to 45 min at 60–75% **HRmax**; **CG**: no interventions	↓**Fat** mass ↓ Testosterone and free testosterone = DHEA, **DHEA-S** =androstenedione
[129]	(*n* = 169) age 50 to 75 years **BMI** ≥ 25 kg/m^2^ (mean 30.4 ± 4.1) Groups: **AG** (*n* = 84) **CG** (*n* = 85)	12 months **AG**: endurance exercise, 5 d/w progressively increase to 45 min at 60–75% **HRmax**; **CG**: no interventions	↓**Fat** mass ↓ Estrone ↓ Estradiol and free estradiol **🡹** **SHBG**
[127]	(*n* = 189) age 50 to 69 years **BMI** 22–40 kg/m^2^ (mean 27.3 ± 3.6) Groups: **AG** (*n* = 96) **CG** (*n* = 93)	12 months **AG**: 3 d/w of combined endurance + strength program (from 60–85% **HRmax**) (2.5 h/w) **CG**: no interventions	↓**Fat** mass =estrogen levels =androgen levels =**SHBG**
[116]	(*n* = 320) age 50–74 years **BMI** 22–40 kg/m^2^ (mean 29.1 ± 4.5) Groups: **AG** (*n* = 160) **CG** (*n* = 160)	12 months **AG**: 225 min/w (average of 3.6 d/w for 178 min/w) at 70% to 80% **HRR** **CG**: no interventions	↓ Estradiol and free estradiol **🡹** **SHBG** ↓ Body weight = estrone, androstenedione and testosterone
[119]	(*n* = 439) age 50–75 years **BMI** ≥ 25 kg/m^2^ (mean 30.9) Groups: **DG** (*n* = 118) **AG** (*n* = 117) **DAG** (*n* = 117) **CG** (*n* = 87)	12 months; **DG**: daily energy intake of 1200 to 2000 kcal/d based on baseline weight; **AG**: ≥45 min **MVPA** (70% to 85% heart rate max), 5 d/w; **DAG**: both interventions; **CG**: no interventions	↓**Fat** mass in all groups vs. **CG** ↓ Insulin in **DG** and **DAG** ↓ **hs-CRP** in **DG** and **DAG** ↓ Leptin in all groups vs. **CG** **🡹** Adiponectin in **DG** and **DAG** ↓ Estron, estradiol, free estradiol, and free testosterone in all in all intervention groups vs. **CG** ↓ Total testosterone in **DAG** **🡹** **SHBG** in **DG** and **DAG**
[126]	(*n* = 439) aged 50–75 years **BMI** ≥ 25 kg/m^2^ (mean 30.9) Groups: **DG** (*n* = 118) **AG** (*n* = 117) **DAG** (*n* = 117) **CG** (*n* = 87)	12 months; **DG**: daily energy intake of 1200–2000 kcal/d based on baseline weight; **AG**: ≥45 min of **MVPA** (70–85% heart rate max), 5 d/w (225 min/w). **DAG**: both interventions (diet + exercise) **CG**: no interventions	↓**Fat** mass in all intervention groups vs. **CG**; ↓ Waist circumference in all intervention groups vs. **CG**;
[117]	(*n* = 382) age 50–74 years **BMI** 22–40 kg/m^2^ (mean 29.4 ± 4.4) Groups: **COG** (*n* = 193) **MVG** (*n* = 189)	12 months of endurance activity (5 d/w, 3 supervised, 2 unsupervised); **COG**: 60 min/session, 60–80% **HRR** **MVG**: 30 min/session, 60–80% **HRR**	↓**Fat** mass depending on exercise volume (high or moderate); =sex hormone levels between groups
[120]	(*n* = 243) age 50–69 years **BMI** 25–35 kg/m^2^ (mean 29.5 ± 2.6) Groups: **DG** (*n* = 97) **COG** (*n* = 98) **CG** (*n* = 48)	16 weeks; **DG**: caloric deficit of 3500 kcal/w with habitual physical activity level; **COG**: 4 h/w of combined endurance (from 60% to 90% **HRR**) and strength program with an average energy expenditure of 2530 kcal/week; **CG**: habitual physical activity level + standardized diet	↓**Fat** mass in **DG** and **COG** ↓ **hs-CRP** in **DG** and **COG**; =IL6 in all groups; ↓ Adiponectin in **COG**; ↓ Leptin in **DG** and **COG**
[121]	(*n* = 41) age 50–74 years **BMI** 23.8–32.9 kg/m^2^ (mean 28.2 ± 3.4) Groups: **AG** (*n* = 22) **CG** (*n* = 19)	6 months; **AG**: 3 d/w progressively increase to 50 min at 70–80% **HRmax**; **CG**: no interventions	=Leptin =Resistin =**Fat** mass **🡹** Aerobic Fitness level ↓ **BMI**
[122]	(*n* = 306) age 50–74 years **BMI** 22–40 kg/m^2^ (mean 29.0) Groups: **AG** (*n* = 153) **CG** (*n* = 154)	12 months; **AG**: 45 min/d, 5 d/w (70% to 80% **HRR**); **CG**: no interventions	↓ Total estradiol =estrogen metabolites and metabolic pathways
[123]	*(n* = 439) age 50–75 years **BMI** ≥ 25 kg/m^2^ (mean 30.0 ± 3.7) Groups: **DG** (*n* = 118) **AG** (*n* = 117) **DAG** (*n* = 117) **CG** (*n* = 87)	12 months + 18 months follow-up (**FU**); **DG**: daily energy intake of 1200–2000 kcal/d based on baseline weight; **AG**: ≥45 min of **MVPA** (70–85% heart rate max), 5 d/w (225 min/w); **DAG**: both interventions (diet + exercise); **CG**: no interventions	**🡹****SHBG** in **DAG**; =**SHBG** in **DG** and **AG**; Participants who reported weight loss had statistically greater decreases in free estradiol, free testosterone, and increases in **SHBG**
[118]	(*n* = 333) age 50–74 years **BMI** 22–40 kg/m^2^ (mean 28.9 ± 4.4) Groups: **COG** (*n* = 168) **MVG** (*n* = 165)	12 months of endurance activity (5 d/w, 3 supervised, 2 unsupervised) + 12 months follow-up (**FU**); **COG**: 60 min/session, 65% to 75% **HRR**; **MVG**: 30 min/session, 65% to 75% **HRR**	↓**Fat** mass depending on exercise volume (high or moderate); ↓ **hs-CRP**, insulin, glucose, **HOMA-IR**, estrone, estradiol, free estradiol at 12 months; **SHBG** at 12 months; ↓ Glucose, insulin, **HOMA-IR**, estrone at **FU**; **🡹** **hs-CRP**, free estradiol, estradiol at **FU**; ↓ **SHBG** at **FU** =biomarker changes over the time between groups.
[124]	(*n* = 35) age 50–65 years **BMI** ≥ 25 kg/m^2^ (mean 33.2 ± 1.4) Groups: **AG** (*n* = 10) **COG** (*n* = 13) **CG** (*n* = 12)	12 weeks, 3 d/w; **AG**: 60 min/session of endurance exercise, 55–75% of **HRR**; **COG**: 40 mn resistance (6 exercises, 3 sets of 8–12 repetition at 65% of **1RM**) + 20 min endurance exercise; **CG**: no interventions	↓**Fat** mass in **AG** and **COG**; ↓ Lean body mass in the **COG**; ↓ **DHEA-S** (−13%), total (−40%) and free testosterone (−41%) in **AG**; ↓ Total (25%) and free testosterone (21%) in **COG**; =estrogen levels in both groups. The decrease in fat mass and **DHEA-S** correlates with an increase in circulating **SHBG**.

Legend: **AG**, aerobic group; **DG**, diet group; **DAG**, diet + aerobic group; **CG**, control group; **COG**, combined group (diet + exercise); **HVG**, high volume group; **MVG**, moderate volume group; **MVPA**, moderate to vigorous physical activity; **HRR**, heart rate reserve; **HRmax**, maximal heart rate; **FU**, follow-up; **w**, week; **d**, days, **HOMA-IR**, fasting glucose (mmol/L) x fasting insulin (mIU/mL)/22.5; **hs-CRP**, high-sensitivity C-reactive protein; **SHBG**, sex-hormone-binding globulin; **DHEA-S**, dehydroepiandrosterone sulfate; **1RM**, one-rep maximum; **BMI**, body mass index. The Red color indicates reported parameters worsened during experimental trails.

**Table 3 ijerph-18-09834-t003:** Exercise and physical activity as secondary and tertiary prevention among ER+ breast cancer survivors before and during pharmacological treatments.

**REF.**	**GROUPS**	**EXERCISE CHARACTERISTICS**	**MAIN OUTCOMES**
[152]	(*n* = 85) age 52–62 years **IRG** (*n* = 42) **DRG** (*n* = 42)	6 months (after treatment) 2 d/w **IRG** and **DRG**: 3 set 12 repetition (13 w supervised + 13 w no-supervised)	↓ Body fat and IGF-II in **IRG** **🡹** **IGFBP-3** in **IRG**
[153]	(*n* = 66) age 46–58 years **HAG** (*n* = 22) HRG (*n* = 21) **CG** (*n* = 23)	6 months (before and during adjuvant **CH**) 4 d/w **HAG**: 15–30 min HRG: 2 sets 10 repetitions **CG**: no intervention	**🡹** Aerobic capacity (25%) in **HAG** and (4%) in HRG ↓ BMD (6.2%) in **CG**, (4.9%) in HRG and (0.7%) in **HAG** ↓ Aerobic capacity (10%) in **CG**
[154]	(*n* = 101) Age **HARG** (*n* = 51) **CG** (*n* = 50)	16 weeks (after **CH**, **RT**, during **HT**) 2 d/w **HARG**: 50-min supervised strength + 90 min unsupervised aerobic **CG**: no intervention	↓ Fast insulin and hip circumference in **HARG** = insulin resistance, fasting glucose and **BMI**
[155]	(*n* = 90) age 41–48 years **DG** (*n* = 29) DHARG (*n* = 29) DHARG + **FVLF** (32)	6 months 5 d/w **DG**: Calcium reach Diet DHARG: Calcium Diet + 150 min of **MVPA** AT + **RT** DHARG + **FVLF**: Calcium Diet + exercise + **FVLF**	↓ Waist circumference and % body fat in DHARG + **FVLF** = insulin, proinsulin, **IGF-1**, **CRP**, cholesterol, **SHBG**, **IL-1B**, and **TNFR2** in all groups **🡹** In QoL in all groups
[156]	(*n* = 75) age 55–64 years **AG** + **HAG** (*n* = 37) **CG** (*n* = 38)	6 months (after **CH**) 5 d/w **AG** + **HAG**: 3 d/w 150 min/week of supervised gym- and 2 d/w home-based moderate-intensity aerobic exercise **CG**: no intervention	↓ FAT in **AG** + **HAG** **🡹** **LM** in **AG** + **HAG** ↓ FAT, **LM** and BMD in **CG**
[157]	(*n* = 26) Age 40–60 years **AG** (*n* = 16) **CG** (*n* = 10)	8 weeks **AG**: moderate intensity **CG**: no intervention	↓ Waist circumference in **AG** **🡹** **PA** level = blood pressure, **HDL**, insulin resistance and **CRP**
[158]	(*n* = 90) age 55–65 years **DARG** (*n* = 47) **CG** (*n* = 43)	6 months (duirng and after **CH**) 3 d/w **DARG**: 30 min 65–80% predicted **HRmax** + 10–15 min resistance band exercise + total daily caloric intake 600 kcal below their requirements **CG**: no intervenation	↓ Central adiposity, WHR, total cholesterol and leptin in **DARG** **🡹** Predicted **VO_2max_** in **DARG**
[159]	(*n* = 28) age 56–66 years **ARG** (*n* = 15) **CG** (*n* = 13)	12 weeks, supervised 6 wks, unsupervised 6 wks **ARG**:, 150 min/wk aerobic moderate intensity + resistance exercise, 2 d/wk **CG**: no intervention	**🡹** IL6 and predicted O_2_ in **ARG** ↓ IL-10, adiponectin, fatigue and sleep disturbance in **ARG**
[160]	(*n* = 46) age 30–70 years **ARG** (*n* = 22) **CG** (*n* = 24)	12 weeks 5 d/w **ARG**: 160 min/wk at 48–52% of heart rate reserve +resistance exercise **CG**: no intervention	↓ %BF, IL10, anxiety, sleep dysfunction, exercise social support in **ARG** **🡹** **VO^2max^** in **ARG**
[161]	(*n* = 242) age ≥ 18 years **AG** (*n* = 78) **RG** (*n* = 82) **CG** (*n* = 82)	3 d/w during **CH** **AG**: 45 min at 80% **VO_2max_** **RG**: two sets of 8–12 at 60–70% of estimated 1-RM GC: no intervention	**🡹****DFS** and **RFI** in **AG** and **RG**
[162]	(*n* = 165) **DG** (*n* = 83) **DRG** (*n* = 82)	12 months (after **CH**, during antiestrogenic treatment) 7 d/w **DG**: food naturally high in proteins, calcium, probiotics and prebiotics **DRG**: diet + 4 reps of 1 isometric exercise	↓ Weight and fat in **DG** and **DRG** ↓ Visceral fat in **DRG**
[163]	(*n* = 36) age 63–75 years **ARG** (*n* = 18) **CG** (*n* = 18)	9 months (during **AI**) 3 d/w **ARG**: 30 min at 75/80% **HRmax** + 3 sets 8–10 reps at 75% **1RM** **CG**: no intervention	**🡹** Osteocalcin in **ARG** ↓ Total mass, total fat and **HDL** in **ARG**
[164]	(*n* = 240) age 52–64 years **ARG**-HIIT (*n* = 79) **AG**-HIIT (*n* = 80) **CG** (*n* = 81)	16 weeks 2 d/w + 12 months **FU** (during and after **CH**) **ARG**-HIIT: 3 sets 10 rep at 70–80% **1RM** + 3 × 3-min bouts on cycle ergometer, 1 min of recovery **AG**-HIIT: from 20 min **MACT** to aerobic part of **ARG**-HIIT **CG**: no intervention	**🡹** Role functioning in **RG**-HIIT and **AG**-HIIT ↓ Total cancer-related fatigue in **RG**-HIIT and **AG**-HIIT **🡹** Pain in **CG**
[69]	(*n* = 23) age 51–63 years **ARG**-HIIT (*n* = 8) **AG**-HIIT (*n* = 9) **CG** (*n* = 13)	16 weeks 2 d/w + 12 months **FU** (during and after **CH**) **ARG**-HIIT: 3 sets 10 rep at 70–80% **1RM** + 3 × 3-min bouts on cycle ergometer, 1 min of recovery **AG**-HIIT: from 20 min **MACT** to aerobic part of **ARG**-HIIT **CG**: no intervention	**🡹** Muscle fiber **CSA** and **SC** count per fiber in **ARG**-HIIT ↓ Symptoms and displayed gains in lower limb in **ARG**-HIIT and **AG**-HIIT **🡹** Number of capillaries per fiber in **AG**-HIIT ↓ **MHC** isoform type I and protein levels of PINK1 in **CG** **🡹** **SOD**2 level in **CG**

Legend: **CH**, chemotherapy; **RT**, radiotherapy; **HT**, hormonal therapy; **AI**, aromatase inhibitor; **AG**, aerobic group; **RG**, resistance group; **HAG**, home-based aerobic group; **ARG**, aerobic + resistance group; **HARG**, home-based aerobic + resistance group; **DG**, diet group; **DRG**, diet + resistance group; **DARG**, diet + aerobic + resistance group; **IRG**, immediate resistance group; **DRG**, delayed resistance group; **CG**, control group; **FVLF**, fruit and vegetable, low-fat diet; **MVPA**, moderate to vigorous physical activity; **MACT** = moderate aerobic continuous training; **VO_2max_**, maximal oxygen consumption; **1RM**, one-rep maximum **BMI**, body mass index; **Fat**, body fat mass; **LM**, lean mass; **ACSM**, American College of Sport Medicine; **HRmax**, maximal heart rate; **FU**, follow-up; **min**, minutes; **DFS**, disease-free survival; **RFI**, recurrence-free interval; **IGFBP-3**, insulin-like growth-factor-binding protein 3; **IGF-1**, insulin-like growth factor; **CRP** = C-reactive protein; **SHBG**, sex-hormone-binding globulin; **SOD**, superoxide dismutase; **MHC**, myosin heavy chain; **CSA**, cross-sectional area; **SC**, satellite cells; **FACT-B**, the functional assessment of cancer therapy—breast; **FACT-G**, the functional assessment of cancer therapy—general; **SF-36**, short form health survey; **TNFR2**, tumor necrosis factor receptor 2; **IL-1B**, interleukin 1 beta; **HDL**, high-density lipoprotein; **BSAP**, bone-specific alkaline phosphatase; **PA**, physical activity.

**Table 4 ijerph-18-09834-t004:** Exercise and physical activity as quaternary prevention in advanced-stage breast cancer patients.

REF.	GROUPS	EXERCISE CHARACTERISTICS	MAIN OUTCOMES
[170]	(*n* = 13) Life expectancy ≥ 6 months age 44–75 years **YG** (*n* = 13)	**YG**: 8 weekly group session (during **CH**)	↓ Pain, fatigue distress **🡹** Relaxation, invigoration
[171]	(*n* = 101) Life expectancy ≥ 12 months age 59–59 years **AG** (*n* = 48) CG (*n* = 53)	16 weeks (during **CH**) **AG**: 150 min MVPA per week CG: no intervention	=min/w exercise and physical functioning in **AG**
[172]	(*n* = 65) age 62–72 years **AG** (*n* = 33) CG (*n* = 32)	12 weeks 3 d/w (during **CH**) **AG**: 55–80 % **VO2peak** on treadmill CG: no intervention	**🡹****VO2peak** and functional capacity in **AG**. Attendance rate 63%, permanent discontinuation 27%, dose modification 49%, acceptable tolerability 42% in **AG**.
[173]	(*n* = 14) Life expectancy at least 4 months age ≥ 18 years **HARG** (*n* = 8) CG (*n* = 6)	8 weeks 2 d/w (during **CH** and **HT**) **HARG**: supervised **RT** 2 sets of 12 repetitions 1 min recovery, intensity 6–7 Adult OMNI Scale + unsupervised 10–15 min walking CG: no intervention	**🡹****FACIT-F** score, **VO_2max_** and six-minute walking test in **HARG**. Adherence 100% in **RT** and 25% in walking.
[174]	(*n* = 48) Life expectancy ≥ 9 months age 56–67 years **YG** (*n* = 30) CG (*n* = 18)	8 week 5–6 d/w (undergoing treatments) **YG**: meditation, gentle postures, breathing techniques, presentations on yogic principles for optimal coping. 15–30 min/d CG: Discussion about several topic related to the disease concerns	↓ Pain levels in **YG** and CG Dose–response relationship between **YG**, duration and daily pain.
[175]	(*n* = 49) age 55–65 years **HPA** (*n* = 49)	6 months (during **CH**, **RT**, **HT**, **TT**) **HPA**: reach 10,000 steps per day.	**🡹****HPA** increases 6-MWT, quadriceps strength ↓ **BMI** =muscle **CSA**, skeletal muscle radiodensity, **LM**.

Legend: **CH**, chemotherapy; **RT**, radiotherapy; **HT**, hormonal therapy; **TT**, targeted therapy; **AG**, aerobic group; **HARG**, home-based aerobic + resistance group; **YG**, yoga group; **HPA**, home physical activity; **CG**, control group; **MVPA**, moderate to vigorous physical activity; **BMI**, body mass index; **VO2peak**, peak oxygen uptake; **FACIT-F**, functional assessment of chronic illness therapy—fatigue; **CSA**, cross-sectional area; **LM**, lean body mass; **6MWT**, 6 min walking test; **min** = minutes.

## Data Availability

No new data were created or analyzed in this study. Data sharing is not applicable to this article.

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
