# Peer review of "Estrogen-Receptor-Positive Breast Cancer in Postmenopausal Women: The Role of Body Composition and Physical Exercise"

_ijerph, 2021, doi:10.3390/ijerph18189834_

Round 1
Reviewer 1 Report
Dear authors, I hope you are enjoying the arrival of August. Thank you for submitting your work.
However, as a reviewer, it is impossible to review the article in the format presented as it does not follow the structure established by the journal. Without presenting a section or paragraph with the methodology followed, it is impossible to assess the quality of the research. Therefore, know your internal consistency and apply PRISMA as necessary in this investigation case.
Author Response
Thank you for your reply regarding our manuscript No. ijerph-1346438 entitled “Estrogen Receptor positive breast cancer in postmenopausal women: the role of body composition and physical exercise”. We are grateful for reviewer’s comments and we appreciate the opportunity that we have been given to further revise and improve our manuscript. All changes were left tracked in the revised text with a note showing the reviewer's request.
REVIEWER 1
Q1. Dear authors, I hope you are enjoying the arrival of August. Thank you for submitting your work.
However, as a reviewer, it is impossible to review the article in the format presented as it does not follow the structure established by the journal. Without presenting a section or paragraph with the methodology followed, it is impossible to assess the quality of the research. Therefore, know your internal consistency and apply PRISMA as necessary in this investigation case.
A1. We thank the Reviewer for the comment, however, we cannot answer to the reviewer’s request because our manuscript is a narrative review and not a systematic review. We verified in the instruction for authors guidelines that the narrative review can use the template file only to prepare the front and back matter of the manuscript. The total research articles structure and PRISMA guidelines should be used only for structured reviews (systematic review) and meta-analyses.
Reviewer 2 Report
In “Estrogen Receptor positive breast cancer in postmenopausal women: the role of body composition and physical exercise”, Dimauro et. al. provides a complete revision of human studies regards the impact of physical activity (PA) or exercise in primary, secondary, tertiary and quaternary prevention of ER positive breast cancer (BC). Given the importance of estrogen receptors, hormones exposure and body composition in BC risk and progression, a summary of the molecular mechanism by which this parameters influence BC is provided as well as the effect of PA in these parameters n clinical interventions.
They clearly summarize the statement of knowledge covering a broad number of clinical studies, I think this review will improve the understanding of this topic in the field of BC
Minnor comments:
1.- In my opinion Figure 1 could be improved for example by showing ER dimerization and also the cell nucleus and cytoplasm.
2.- Line 103, in my opinion the sentence “Probably the uncertainty about the role of ERβ is imputable to the presence of five differ-103 ent isoforms of Erβ (Erβ, β2, β3, β4, and β5), which makes it more difficult to recognize its involvement in BC carcinogenesis.“ should be modified. Although more investigations are needed to determine the precise role of ER beta in BC, the general consensus is that has a suppressor role in BC (doi: 10.1186/s13046-019-1359-9, doi: 10.1016/j.steroids.2008.04.006., doi: 10.3389/fendo.2018.00781,
3.- To demonstrate the plasticity of adipose tissue, in line 165, where says “According to the anatomical location and to the main cell component, adipose tissue 165 can be divided in three different types: white adipose tissue (WAT), which represents 166 more than 95% of the fat mass, brown adipose tissue (BAT), which constitutes 1% to 2% 167 of fat, and beige adipose tissue, more difficult to quantify and capable to switch into 168 brown-like adipocytes upon exposure to either cold or adrenergic stimulation [39].“, authors could also add that conversion of white AT to beige AT and the conversion of brown AT to beige has been also observed, may reviews summarize this.
4.- To abbreviate C-reactive protein authos should use the oficial abreviation CRP, instead of CPR.
Author Response
Thank you for your reply regarding our manuscript No. ijerph-1346438 entitled “Estrogen Receptor positive breast cancer in postmenopausal women: the role of body composition and physical exercise”. We are grateful for reviewer’s comments and we appreciate the opportunity that we have been given to further revise and improve our manuscript. All changes were left tracked in the revised text with a note showing the reviewer's request.
REVIEWER 2
Minor comments:
Q1. In my opinion Figure 1 could be improved for example by showing ER dimerization and also the cell nucleus and cytoplasm.
A1. As requested by the Reviewer we changed and improved the figure1. In particular, we added the receptor dimerization, and the cellular compartmentalization (i.e., nucleus and cytoplasm). Furthermore, we revised the figure legend 1, accordingly.
Q2. Line 103, in my opinion the sentence “Probably the uncertainty about the role of ERβ is imputable to the presence of five different isoforms of Erβ (Erβ, β2, β3, β4, and β5), which makes it more difficult to recognize its involvement in BC carcinogenesis.“ should be modified. Although more investigations are needed to determine the precise role of ER beta in BC, the general consensus is that has a suppressor role in BC (doi: 10.1186/s13046-019-1359-9, doi: 10.1016/j.steroids.2008.04.006., doi: 10.3389/fendo.2018.00781).
A2. We thanks the Reviewer for this proper comment. Different studies reported an anti-cancer role of ERβ in BC. However the mechanisms by which the receptor exerts these effects are not totally clear. To better describe this concept, as suggested by the Reviewer, we changed the sentence in the revised manuscript. The following references have been added:
- Fox EM, Davis RJ, Shupnik MA. ERbeta in breast cancer--onlooker, passive player, or active protector? Steroids. 2008 Oct;73(11):1039-51. doi: 10.1016/j.steroids.2008.04.006. Epub 2008 Apr 20. PMID: 18501937; PMCID: PMC2583259.;
- Song, P., Li, Y., Dong, Y. et al.Estrogen receptor β inhibits breast cancer cells migration and invasion through CLDN6-mediated autophagy. J Exp Clin Cancer Res 38, 354 (2019).
Q3. To demonstrate the plasticity of adipose tissue, in line 165, where says “According to the anatomical location and to the main cell component, adipose tissue can be divided in three different types: white adipose tissue (WAT), which represents more than 95% of the fat mass, brown adipose tissue (BAT), which constitutes 1% to 2% of fat, and beige adipose tissue, more difficult to quantify and capable to switch into brown-like adipocytes upon exposure to either cold or adrenergic stimulation [39].“, authors could also add that conversion of white AT to beige AT and the conversion of brown AT to beige has been also observed, many reviews summarize this.
A3. We thanks the Reviewer for this comment. As suggested we have better explored the topic of adipose tissue conversion and modified the sentence in the revised manuscript.
The following References have been added:
- Zorena, K.; Jachimowicz-Duda, O.; Ślęzak, D.; Robakowska, M.; Mrugacz, M. Adipokines and Obesity. Potential Link to Metabolic Disorders and Chronic Complications. J. Mol. Sci.2020, 21, 3570
- Kaisanlahti A, Glumoff T. Browning of white fat: agents and implications for beige adipose tissue to type 2 diabetes. J Physiol Biochem. 2019 Feb;75(1):1-10.
- Mulya A, Kirwan JP. Brown and Beige Adipose Tissue: Therapy for Obesity and Its Comorbidities? Endocrinol Metab Clin North Am. 2016 Sep;45(3):605-21.
Q4.To abbreviate C-reactive protein authors should use the official abbreviation CRP, instead of CPR.
A4. The abbreviation CPR referred to C-reactive protein has been changed with CRP.
Reviewer 3 Report
this work brings to the attention a topic that I consider very interesting, that is the practice of physical exercise in relation to cancer, I absolutely agree on the need to bring to the attention of those who manage public health, the importance of providing such an approach.
However, some points should be revised:
- certainly, a careful revision of the style must be made, possibly by a mother tongue, many periods are unclear and difficult to read.
- in the conclusions, but also in the work, there is no direct link between the estrogen receptor and physical exercise.
- it should be reported if physical activity was practiced even before the diagnosis and/or therapy of the tumor, this has a lot of influence on the effect of the physical activity itself
- a training scheme should also be proposed for both aerobic and strength, perhaps combining the two, which seems the most effective strategy, even better if in relation to the type of prevention, the type of tumor, and perhaps the body composition of the subjects
The present work ref 39 and 40 should be better chosen on the classification of different types of adipose tissue
line 174 inflammatory cells? Please characterize better
line 183 please rewrite the phrase
line 187-200 should be better characterized especially involving the lowgrade inflammation concept
line 205 very hard to read
line 213-220 is an interesting point but it should be better reported, it is funny to read fit for fat cell, but it should be better reported
line 223 "normal weight" is not scientific, you should consider BMI or body fat percentage
line 360 it is hard to imagine an increase of muscular mass in a hypocaloric regimen, is there a reference?
Author Response
Thank you for your reply regarding our manuscript No. ijerph-1346438 entitled “Estrogen Receptor positive breast cancer in postmenopausal women: the role of body composition and physical exercise”. We are grateful for reviewer’s comments and we appreciate the opportunity that we have been given to further revise and improve our manuscript. All changes were left tracked in the revised text with a note showing the reviewer's request.
REVIEWER 3
Q1. Certainly, a careful revision of the style must be made, possibly by a mother tongue, many periods are unclear and difficult to read.
A1. We apologize for the language errors. As suggested by the Reviewer the manuscript has been totally revised and edited by an English native speaker.
Q2. In the conclusions, but also in the work, there is no direct link between the estrogen receptor and physical exercise.
A2. Thanks to reviewer for this interesting comment that allowed us to better clarify this aspect. All the studies mentioned in the chapter 5 as well as in tables 1-4 “Physical activity and exercise as fundamental approaches within ER positive breast cancer at each disease prevention levels” reported the main outcomes derived from the analysis of patient sera, thus unfortunately they did not investigate the ER modulation. To date, few pre-clinical animal studies demonstrated an effect of PA in reducing ER expression and in promoting apoptosis in mammary gland cells isolated from rats affected by breast cancer. To better clarify this concept, we added a sentence at the end of 2nd chapter and revised a sentence in the conclusion session.
The following new References have been reported:
- Wang M, Yu B, Westerlind K, Strange R, Khan G, Patil D, Boeneman K, Hilakivi-Clarke L. Prepubertal physical activity up-regulates estrogen receptor beta, BRCA1 and p53 mRNA expression in the rat mammary gland. Breast Cancer Res Treat. 2009 May;115(1):213-20. doi: 10.1007/s10549-008-0062-x. Epub 2008 May 31. PMID: 18516675; PMCID: PMC4474161;
- Siewierska K, Malicka I, Kobierzycki C, Grzegrzolka J, Piotrowska A, Paslawska U, Cegielski M, Podhorska-Okolow M, Dziegiel P, Wozniewski M. Effect of Physical Training on the Levels of Sex Hormones and the Expression of Their Receptors in Rats With Induced Mammary Cancer in Secondary Prevention Model - Preliminary Study. In Vivo. 2020 Mar-Apr;34(2):495-501.
- Shelley M. Enger, Ronald K. Ross, Annlia Paganini-Hill, Catherine L. Carpenter and Leslie Bernstein
Body Size, Physical Activity, and Breast Cancer Hormone Receptor Status: Results from Two Case-Control Studies Cancer Epidemiol Biomarkers Prev July 1 2000 (9) (7) 681-687
Q3. It should be reported if physical activity was practiced even before the diagnosis and/or therapy of the tumor, this has a lot of influence on the effect of the physical activity itself
A3. As suggested by Reviewer, we included a sentence at the end of 5th chapter, highlighting the main characteristics of BC patients reported in all research articles included in our tables
We added a new Reference:
- Ainsworth BE, Haskell WL, Herrmann SD, Meckes N, Bassett DR Jr, Tudor-Locke C, Greer JL, Vezina J, Whitt-Glover MC, Leon AS: Compendium of physical activities: a second update of codes and MET values. Med Sci Sports Exerc 2011, 2011(43):1575–1581.
Q4. A training scheme should also be proposed for both aerobic and strength, perhaps combining the two, which seems the most effective strategy, even better if in relation to the type of prevention, the type of tumor, and perhaps the body composition of the subjects
A4 We thanks the Reviewer for this comment because it allow us to clarify this important aspect. The definition of a detailed training protocol is generally difficult to define without to considerer the characteristics of subjects. To date, considering the complexity of this pathology it is still more difficult to delineate in detail the physical activity protocols. We believe it is more useful to indicate, on the basis of numerous scientific researches, some "general" guidelines to be used as a starting point for designing a tailored and suitable motor activity protocol considering the characteristics of each individual patient. To this aim, we had elaborated the Figure 3 already including the main recommendations to follow to design a personalized exercise program.
Q5. The present work ref 39 and 40 should be better chosen on the classification of different types of adipose tissue
A5. As suggested by the Reviewer we added new References:
- Zorena, K.; Jachimowicz-Duda, O.; Ślęzak, D.; Robakowska, M.; Mrugacz, M. Adipokines and Obesity. Potential Link to Metabolic Disorders and Chronic Complications. J. Mol. Sci.2020, 21, 3570
- Kaisanlahti A, Glumoff T. Browning of white fat: agents and implications for beige adipose tissue to type 2 J Physiol Biochem. 2019 Feb;75(1):1-10.
- Mulya A, Kirwan JP. Brown and Beige Adipose Tissue: Therapy for Obesity and Its Comorbidities? Endocrinol Metab Clin North Am. 2016 Sep;45(3):605-21.
- Muller, S., Ader, I., Creff, J. et al.Human adipose stromal-vascular fraction self-organizes to form vascularized adipose tissue in 3D cultures. Sci Rep 9, 7250 (2019).
Q6. line 174 inflammatory cells? Please characterize better
A6. We thanks the Reviewer for the proper question, and we apologize for the confusing expression. We revised the sentence.
The new reference has been added
- Huh JY, Park YJ, Ham M, Kim JB. Crosstalk between adipocytes and immune cells in adipose tissue inflammation and metabolic dysregulation in obesity. Mol Cells. 2014 May;37(5):365-71. doi: 10.14348/molcells.2014.0074.
Q7. line 183 please rewrite the phrase
A7. The sentence has been modified in the revised manuscript. New reference has been added:
- Margot P. Cleary, Michael E. Grossmann, Obesity and Breast Cancer: The Estrogen Connection, Endocrinology, Volume 150, Issue 6, 1 June 2009, Pages 2537–2542
Q8. line 187-200 should be better characterized especially involving the lowgrade inflammation concept A8. As suggested by the Reviewer we modified and better discuss the paragraph in the revised manuscript. New References have been added:
- Chawla A, Nguyen KD, Goh YP. Macrophage-mediated inflammation in metabolic disease. Nat Rev Immunol. 2011 Oct 10;11(11):738-49. doi: 10.1038/nri3071. PMID: 21984069; PMCID: PMC3383854.
- Burhans MS, Hagman DK, Kuzma JN, Schmidt KA, Kratz M. Contribution of Adipose Tissue Inflammation to the Development of Type 2 Diabetes Mellitus. Compr Physiol. 2018 Dec 13;9(1):1-58. doi: 10.1002/cphy.c170040. PMID: 30549014; PMCID: PMC6557583.
Q9. line 205 very hard to read.
A9. The sentence has been modified and improved in the revised manuscript. New references have been added:
- Sgrò P, Emerenziani GP, Antinozzi C, Sacchetti M, Di Luigi L. Exercise as a drug for glucose management and prevention in type 2 diabetes mellitus. Curr Opin Pharmacol. 2021 Aug;59:95-102.
- Birbrair A, Zhang T, Wang ZM, Messi ML, Enikolopov GN, Mintz A, Delbono O. Role of pericytes in skeletal muscle regeneration and fat accumulation. Stem Cells Dev. 2013 Aug 15;22(16):2298-314.
- Lafontan M, Langin D. Lipolysis and lipid mobilization in human adipose tissue. Prog Lipid Res. 2009 Sep;48(5):275-97.
- Jaworski K, Sarkadi-Nagy E, Duncan RE, Ahmadian M, Sul HS. Regulation of triglyceride metabolism. IV. Hormonal regulation of lipolysis in adipose tissue. Am J Physiol Gastrointest Liver Physiol. 2007 Jul;293(1):G1-4.
Q10. line 213-220 is an interesting point but it should be better reported, it is funny to read fit for fat cell, but it should be better reported
A10. Thanks to Reviewer for the comment. We clarify the meaning of “fit” expression in the revised manuscript.
Q11. line 223 "normal weight" is not scientific, you should consider BMI or body fat percentage.
A11. We modified the sentence in the revised manuscript considering the percentage of BF according the article “Dympna Gallagher, Steven B Heymsfield, Moonseong Heo, Susan A Jebb, Peter R Murgatroyd, Yoichi Sakamoto, Healthy percentage body fat ranges: an approach for developing guidelines based on body mass index, The American Journal of Clinical Nutrition, Volume 72, Issue 3, September 2000, Pages 694–701, https://doi.org/10.1093/ajcn/72.3.694”
Q12. line 360 it is hard to imagine an increase of muscular mass in a hypocaloric regimen, is there a reference?
A12. Thanks to the Reviewer for this comment. It is known that the weight loss achieved through a hypocaloric regimen, decreases both fat and fat-free mass. However, different studies suggest that regular physical activity associated with an high protein intake (1.25–1.5 times the RDA for sedentary persons and >1.5 times the RDA for those who exercise) could be useful to preserve as much as possible the loss of muscle mass especially for persons with obesity who undergo weight-loss therapy. We apologize for this misunderstanding and to better describe this concept we revised the sentence in our manuscript.
New References have been added:
- Thomas DT, Erdman KA, Burke LM. American College of Sports Medicine Joint Position Statement. Nutrition and Athletic Performance. Med Sci Sports Exerc. 2016 Mar;48(3):543-68.
- Wycherley TP, Moran LJ, Clifton PM, Noakes M, Brinkworth GD. Effects of energy-restricted high-protein, low-fat compared with standard-protein, low-fat diets: a meta-analysis of randomized controlled trials. Am J Clin Nutr. 2012 Dec;96(6):1281-98.
- Leidy HJ, Clifton PM, Astrup A, Wycherley TP, Westerterp-Plantenga MS, Luscombe-Marsh ND, Woods SC, Mattes RD. The role of protein in weight loss and maintenance. Am J Clin Nutr. 2015 Jun;101(6):1320S-1329S. doi: 10.3945/ajcn.114.084038. Epub 2015 Apr 29. PMID: 25926512.
Round 2
Reviewer 1 Report
Dear authors, I hope you are enjoying this week.
On the one hand, unfortunately, I do not have a response letter on the comments in the first review, which makes it difficult to know what improvements they made in response to them.
On the other hand, the article lacks a methodology statement, only mentioning that it is a review. Furthermore, they do not state the type of review, what guide they followed for its preparation or reference author, or the inclusion or exclusion criteria and databases.
Although it presents improvements in the presentation of the results, the lack of methodology makes it impossible to review it without knowing how it was arrived at and, above all, to review its internal coherence.
Author Response
A1. We are surprised and really apologized for this comment because we had replied to his/her comment through the Response to Reviewer (Round 1). Nevertheless, as suggested by Journal Guidelines (https://www.mdpi.com/journal/ijerph/instructions#preparation), a Narrative Review manuscript should comprise the ”Front matter” (title, author list, affiliation, abstract, keywords), literature review sections and “Back matter” (Supplementary Materials, Acknowledgments, Author Contributions, Conflicts of Interest, References). The template file can also be used to prepare the front and back matter of your review manuscript. It is not necessary to follow the remaining structure.
Otherwise, Structured Reviews and meta-analyses should use the same structure as research articles and ensure they conform to the PRISMA guidelines, including a dedicated methodology section.
To better clarify this aspect, we revised our methodology statement and specify the type of Review at the end of the Introduction section. Each changing has been left in green.
Reviewer 3 Report
Authors improved the manuscript follow the suggestions.
Author Response
We are grateful for reviewer’s comments and we appreciate the opportunity that we have been given to further revise and improve our manuscript